# ADVL: ADAPTIVE DISTILLATION FOR VISION-LANGUAGE TASKS

## ABSTRACT

Large-scale image-text pairs, such as image-captions and image-phrases, enable the strong representation of vision-language (VL) models. Nevertheless, they lose diversity and complexity due to the constraints in collecting data. Meanwhile, models pre-trained with image-only or text-only data (we call them unimodal pre-trained models) continue to flourish and impress the community. Compared to image-text pairs, unimodal data has less constraints during the collection process resulting in more diverse styles. A natural question is how to leverage unimodal pre-trained models to benefit downstream VL tasks? Most existing works focus on fusing VL information in the expensive pre-training stage. They directly plug in unimodal pre-trained encoders into a VL framework and redo an additional pre-training step on paired image-text data. This causes additional computation expense and the unimodal pre-trained knowledge might be forgotten. In this paper, we take a different route and investigate how to fuse VL information in the finetuning stage only. To directly transfer pre-trained knowledge from unimodal models to help downstream VL tasks, we propose ADVL, which avoids redoing any pre-training step and is generalizable to be applied on top of various VL models. To comprehensively demonstrate the effectiveness of ADVL, we conduct evaluation across three mostly recognized highly semantic VL benchmarks: VCR, VQA, and SNLI-VE under three settings, low-shot, full-shot and domain-shifted settings. Results show that ADVL consistently improves the performance with different VL base models across all settings. It even achieves state-of-the-art (SOTA) performance on VCR among models pre-trained with image-text data and delivers competitive results on VQA and SNLI-VE. Based on our analysis, we also discover that ADVL can improve the robustness of VL models and regulate them to better use vision information.

## 1 INTRODUCTION

Recently, Vision-Language (VL) models (Radford et al., 2021; Jia et al., 2021b; Pham et al., 2021; Wang et al., 2021; Chen et al., 2020; Su et al., 2020; Gan et al., 2020) pre-trained on paired image-text data have achieved great success on many VL tasks (Zellers et al., 2019; Xie et al., 2019; Antol et al., 2015; Deng et al., 2009). These paired data generally fall into two categories: curated image-caption (Sharma et al., 2018; Lin et al., 2014) or noisy online image-text data (Radford et al., 2021; Jia et al., 2021b; Pham et al., 2021; Wang et al., 2021; Yu et al., 2022).

However, primarily using paired image-text data for pre-training restricts the knowledge the model can learn. Image-text data are challenging to collect and their styles are limited. Image captions are typically short, template-like and with limited vocabulary (Chen et al., 2015; Sharma et al., 2018). The left part of Fig. 1 shows an example of image captions, "A glass of beer on a table". This description differs from texts used in many downstream applications, like VCR (Zellers et al., 2019) and SNLI-VE (Xie et al., 2019). For these tasks, event inference, spatial/temporal scene understanding are essential.

On the other hand, unimodal data, such as text, are relatively easier to collect and they cover a wider range of domains. For example, web-crawled text corporas (Raffel et al., 2020; Aaron Gokaslan*) contain longer sentences, paragraphs, stories, and articles with full context. In fact, there exist many readily-available unimodal models pre-trained on billion-scale data.

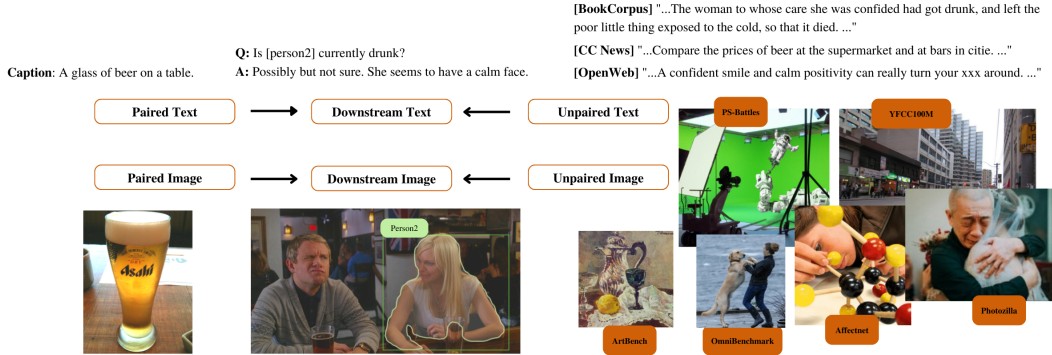

Figure 1: Examples of paired, unpaired and downstream image and text data. Unpaired text and image have more diverse distribution in terms of contents and structures than paired data.

Considering this, we posit that *the diverse styles and sources of text and image data in unimodal pre-training can benefit downstream VL tasks, in addition to paired image-text data.* Inspired by this, we study how to effectively and efficiently transfer pre-trained knowledge from unimodal encoders to improve downstream VL models.

In this paper, we propose ADVL, **A**daptive **D**istillation that leverages unimodal pre-trained models for improving performance on **VL** tasks. ADVL takes unimodal encoders as the teacher models and distill their knowledge into a student VL model that pre-trained on multimodal data. To allow distillation from vision encoders and text encoders from different pre-training sources, ADVL includes separate distillation pathways for vision and language. Further, two adaptive mechanisms, Adaptive Confidence-based Weighting and Adaptive Text Token Selection, are proposed in instance-level and token-level to dynamically adjust the significance of the distilled knowledge. Finally, an adaptive two-step finetuning strategy is introduced to mitigate the domain gap in the distillation.

Prior works (Shen et al., 2021; Tan & Bansal, 2019; Li et al., 2019) has proposed to initialize weights from unimodal vision or text encoders when pre-training the VL models with paired image-text data. Some other studies (Li et al., 2020; Zhou et al., 2022) have focused on joint pre-training with both multimodal and unimodal data. As opposed to these works, ADVL leverages unimodal pre-trained models when fine-tuning a VL model for a downstream task. This setting has the following benefits: (1) *Computation and energy efficient*: pre-training VL models is resource-hungry and may not be practical when computation resource is limited. ADVL avoids redoing the pre-training step. (2) *Modularization and future proof*: ADVL allows researchers to integrate new pre-trained vision, text and multimodal encoders when they become available. This is essential as more and more large unimodal and multimodal pre-trained models are released. (3) *Flexibility and generalization*: The flexibility of selecting various combinations of pre-trained encoders as teacher models enables researchers to explore and find the optimized teacher models (e.g., in the same domain) for a specific downstream task.

To verify the effectiveness of ADVL, we implement ADVL on top of several high-performing VL models and evaluate the results on 3 popular VL tasks, VCR (Zellers et al., 2019), SNLI-VE (Xie et al., 2019), and VQA (Antol et al., 2015), under low-shot, full-data, and domain-shifted settings. With extensive experiments, we show that ADVL improves the performance of VL models on various tasks across all settings by levering unimodel encoders. Moreover, it achieves SOTA performance on VCR-Q2A compared to other non-ensembled models pre-trained with large image-text data. It also achieves competitive performance on SNLI-VE and VQA. Furthermore, we discover the existing VL models tend to under-utilize vision information. With ADVL, the vision information can be better used. As a result, the model is more robust in the domain-shifted setting.

We plan to release the code to facilitate future research in the community.

## 2 RELATED WORK

**Pre-training with image and text data:** Existing pre-training frameworks leveraging image and text data can fall into three essential categories: (1) Unimodal pre-training including image-only pre-training, *e.g.,* (Dosovitskiy et al., 2020; Liu et al., 2021b; Dai et al., 2021) or text-only pre-

training *e.g.,* (Devlin et al., 2018; Raffel et al., 2020; Radford et al., 2019b;a; Liu et al., 2019b); (2) VL pre-training with vision and text encoders, *e.g.,* (Shen et al., 2021; Tan & Bansal, 2019; Li et al., 2019; Radford et al., 2021; Mu et al., 2021; Li et al., 2021b; Jia et al., 2021a; Pham et al., 2021; Yu et al., 2022). Although they are pre-trained with paired image-text data, their structures incorporate independent vision and text encoders. For example, CLIP (Radford et al., 2021) has one vision encoder and one text encoder loosely connected by a contrastive loss in the end. Among them, frameworks like (Shen et al., 2021; Tan & Bansal, 2019; Li et al., 2019) utilize the unimodal pre-trained encoders by directly plugging them into the VL frameworks. Following, they have to conduct an additional step of pre-training on image-caption data before finetuning for downstream tasks. (3)VL pre-training with cross-modal structures only, *e.g.,* (Su et al., 2020; Chen et al., 2020; Gan et al., 2020; Zhang et al., 2021; Wang et al., 2021). They enable in-depth integration between vision and language information in the pre-training stage and thus often achieve leading performance in highly semantic VL benchmarks like VCR.

*In this work*, we emphasize utilizing the learned knowledge from unimodal pre-trained encoders like RoBERTa (Liu et al., 2019b) and ViT (Dosovitskiy et al., 2020). We would further leverage the pre-trained vision and text encoders from VL pre-training like CLIP (Radford et al., 2021) for the comparison purpose among experiments.

**Knowledge Distillation:** Former works use knowledge distillation for model compression (Buciluǎ et al., 2006; Hinton et al., 2015; Wang & Yoon, 2021; Xu et al., 2020; Liu et al., 2019a). Among them, Sanh et al. (2019); Jiao et al. (2019); Sun et al. (2020b) focus on compressing language models and Tian et al. (2020); Fang et al. (2021b) focus on vision models. Past methods leverage various representations for transferring embeded knowledge, *e.g.* logit distribution, intermediate feature embeddings and attentions (Kim & Rush, 2016; Sun et al., 2019; Liu et al., 2020; Yang et al., 2020; Chen et al., 2021). Other methods, Tian et al. (2019); Sun et al. (2020a) also try to create more robust distillation mechanisms like contrastive distillation. Li et al. (2021a); Niu & Zhang (2021); Wu et al. (2021); Furlanello et al. (2018); Kang et al. (2021); Xiang et al. (2020) create different dynamic and adaptive distillation methods like adjusting distillation weights. Related, our method pushes further to propose adaptive mechanisms like instance-level confidence-paced weighting and token-level text-token selection. Together, they help adjust the distillation weights among text tokens and teacher models. With the popularity of VL learning, recent methods investigate on transferring knowledge between modalities (Jin et al., 2022; Gupta et al., 2016). Prior works (Do et al., 2019; Fang et al., 2021a; Mun et al., 2018; Huang et al., 2022) also experiment with transfering VL knowledge between models. However, their approaches do not separate distillation procedures between modalities thus they require both the teacher and student models to be VL models with similar structures. Differently, ADVL incorporates independent distillation procedures for both vision and language, and this allows it to distill knowledge from different pre-trained models. Further, Tang et al. (2021); Liu et al. (2021a); Khan et al. (2021); Croitoru et al. (2021) exam distillation between a single-modal model and a VL framework. Similarly, concurrent works (Zhang et al., 2022; Dai et al., 2022; Gu et al., 2021) explore distilling knowledge from the single-modal encoders within VL contrastive pre-training frameworks. However, they both can utilize at-most one single-modal model as the teacher during distillation. In contrast, our method can simultaneously distill knowledge from vision and text encoders separately as the dual-teachers. ADVL is generalizable as it not only can distill from unimodal pre-trained encoders like RoBERTa (Liu et al., 2019b) but also from vision or text encoders within VL pre-training frameworks like CLIP (Radford et al., 2021).

To our best knowledge, ADVL is the first approach to distill knowledge from mulitple unimodal pre-trained encoders to assist downstream VL tasks. We also conduct the first study on the challenging low-shot evaluation setting for VCR and SNLI-VE. See A.4.2 for additional evaluation and discussion.

## 3 METHOD

**Problem Definition:** We assume two pre-trained teacher models, $f_{t,v}$ and $f_{t,l}$ and a pre-trained student model $f_s$. Our goal is to distill the knowledge from the teacher models to the student on a specific task $D$ with input image $i$, and text $t$.

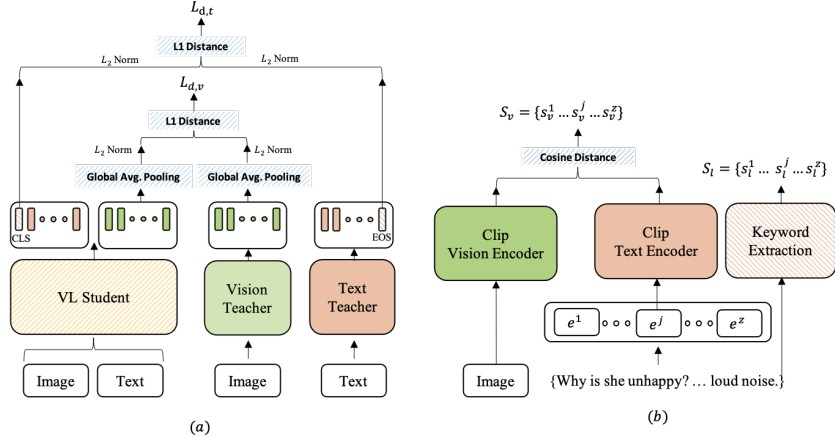

Figure 2: Structure diagrams of Modality-specific Distillation (a) and Text Token Selection (b). On the left diagram (a), within ADVL, every VL model as the student would have a vision teacher and a text teacher to conduct knowledge distillation separately. One the right diagram (b), the text would be pre-processed via tokenization and stopwords removal before input into the Clip text encoder. The keyword extraction is conducted by an off-shelf keyword extractor.

**Overview:** In the following, we first introduce distillation in transformer-based models and how ADVL separates distillation pathways for vision and language. Then, we discuss the adaptive mechanisms of ADVL. Finally, we introduce the adaptive two-step finetuning strategy that further helps VL models to adapt the unimodal pre-trained knowledge gradually in the finetuning stage. On both of the finetuning steps, all the adaptive mechanisms of ADVL are applied at the same time.

### 3.1 MODALITY-SPECIFIC DISTILLATION

First, we identify an attention layer in the student VL model and an attention layer in the teacher model. Then, we conduct knowledge distillation between them. Typically, this would be between the last layers of the teacher and the student model (Sanh et al., 2019; Jiao et al., 2019; Kim & Rush, 2016; Sun et al., 2019). Different from former methods that do not differentiate distillation process between modalities, we propose to separate the distillation of vision and language information so that we can flexibly distill knowledge from various pre-trained vision and text encoders from different sources.

For language distillation, in conventional methods (Sanh et al., 2019; Jiao et al., 2019; Kim & Rush, 2016; Sun et al., 2019), the hidden features of the leading classification token, [CLS], are commonly extracted and used for predicting final labels. Similarly, on either the teacher or student model, we also extract the [CLS] token features from the identified attention layer to represent the information of text sequence. As for vision distillation, in the target distillation layer of either a teacher or a student model, following Lu et al. (2019), we, on purpose, globally average-pool across features of all the visual tokens to create global representative visual features to represent the visual sequence.

After extracting the representative text and visual features, we first normalize them by dividing them by the L2 norm and then calculate the feature distance separately for the two modalities via L1 measure to obtain the corresponding distillation losses. For the text modality, we have

$$L_{d,t} = \left\| \frac{f_{t,l}(t)}{\|f_{t,l}(t)\|_2} - \frac{f_{s,l}(t)}{\|f_{s,l}(t)\|_2} \right\|_1,  \tag{1}$$

where $t$ refers to the input text; $f_{t,l}$ and $f_{s,l}$ refer to the feature extraction of the representative text features in the teacher and student model; Similarly, when the teacher model is a vision encoder, then we can define $L_{d,v}$ accordingly.

Figure 3: Token-level attention values before and after distillation. The top and bottom rows visualilze aggregated token-level attention values of heads from VL-BERT before and after distillation. The middle row indicates the emphasized and selected tokens by using language prior.

## 3.2 ADAPTIVE MECHANISM

### 3.2.1 ADAPTIVE CONFIDENCE-BASED WEIGHTING

As the VL student model improves, it may become more confident and accurate in generating predictions for certain instances than the teacher models. With those instances, we prefer less interference from the teacher models. Considering this, we design a confidence-based weighting mechanism to allow the student model to adjust the distillation weight based on its own confidence.

For each input instance, we first extract the logit vector, $l^s$ output by the student model. We then compute the probability distribution over all classes, $\sigma(l^s)$, by applying a softmax function, $\sigma$, on top. The class with the highest probability is selected and the probability can be regarded as the highest class confidence score, $C = \max(\sigma(l^s))$, of the model's prediction for that instance.

Inspired by Xiang et al. (2020), we further evaluate the student model in the validation set at the end of each training epoch and record the model's highest class confidence score when the student model predicts correctly. Based on these scores, we calculate the average class confidence score, $C_{avg}$ when the student makes the correct prediction. We can regard $C_{avg}$ as a reference value with which the student is confident and accurate in its predictions.

During training, for each input instance, we also calculate the student model's highest class confidence score, $C$. Comparing it against the pre-computed reference score, $\frac{C_{avg}}{C}$, can tell us how much the student is confident about the classification prediction for that instance. As we train the student model from epoch to epoch, the reference score, $C_{avg}$ will also be updated accordingly to reflect the student's confidence in accurate classification predictions. Considering the student model may have low confidence at the initial epochs, we also set a minimum value, 0.5 for the reference score. Following, we utilize the ratio in the equation below to adjust the distillation weight during training:

$$w = \frac{1}{T} \cdot f_{sigmoid}\left(\frac{\max(C_{avg}, 0.5)}{C} - 1\right),\qquad(2)$$

where $f_{sigmoid}$ represents the sigmoid function. $T$ is a learnable temperature value. We directly compare the ratio against 1 via subtraction to determine how confident the student model is for an input instance. If the resulting value is smaller than 0, the student is confident about the prediction on the instance. Hence the overall distillation weight would be decreased accordingly. Otherwise, the student is considered less confident than usual. Then the student may need more help from the teacher model via distillation for that specific instance.

Formally, the distillation loss would then be:

$$L_d = w \cdot (E \cdot L_{d,v} + F \cdot L_{d,t}),\qquad(3)$$

where $E$ and $F$ are learnable weights for vision and language distillation respectively.

### 3.2.2 ADAPTIVE TEXT TOKEN SELECTION

Former works (Ye & Kovashka, 2021; Cao et al., 2020; Ramakrishnan et al., 2018; Kovaleva et al., 2019) point out V+L models are prone to over-rely on trivial information within text data. For instance, for an input text sequence, existing models still struggle in differentiating semantically important text tokens from the trivial ones. As an example in Fig. 3, among the input text sequence, tokens like "person4","doing", "sending" and "telegram" are semantically important since they contain crucial information for understanding the visual scene and answering the question. Yet, from

the top row of visualized attention aggregated over heads from VL-BERT, we can observe that it mistakenly focuses on trivial tokens like "is" and "[SEP]" and neglects the important tokens. This may further result in failure to understand the visual scene. To resolve this, we intend to incorporate language priors to help differentiate semantically important tokens in every input text sequence. As in the second row of Fig 3, relevant tokens are emphasized and selected with language prior. Consequently, we can further utilize the selected tokens by guiding the knowledge distillation procedure in a more refined token-level. This may help the student model in learning the semantic information of the selected tokens instead of shortcuts or biases from the trivial tokens, as shown in the third row of Fig. 3.

To achieve it, we measure the semantic importance of each token based on the input image and text. Specifically, we generate the semantic importance score $s^j$, for each token $e^j, j \in [1, z]$ based on its visual relevance against the image and syntactic importance within the text sequence. Therefore, every $s^j$ is a summation of two subsequent scores, visual relevance score, $s_v^j$ and syntactic importance score, $s_l^j$. We first apply a pre-trained CLIP model (Radford et al., 2021) to measure the visual relevance score between every token and the input image. Thereby, $S_v = \{s_v^j\}_{j=1}^z = \{cos \langle f_{c,v}(v), f_{c,l}(e^j) \rangle, j \in [1, z]\}$. Here, $f_{c,v}$ represents the global feature extraction by CLIP's visual encoder and $f_{c,l}$ denotes output features of local tokens from the text encoder. The cosine distance between the extracted visual and text features would be served as the visual relevance score. Further, we also adapt a pre-trained off-shelf keyword extraction model (Campos et al., 2020) to produce the syntactic importance score, $S_l = \{s_l^j\}_{j=1}^z$. After summation, $S = \frac{S_v}{|S_v|_1} + \frac{S_l}{|S_l|_1}$ We then rank text tokens based on $S$ and select $m$ tokens (A.1.1) with the highest scores, $\mathbf{e} = \{e^i\}_{i=1}^m$. Their corresponding features of both teacher and student models would be then compared with an L1 measure to calculate their difference:

$$L_{d,t}' = \sum_{i=1}^m \left\| \frac{f_{t,l}(e^i)}{\|f_{t,l}(e^i)\|_2} - \frac{f_{s,l}(e^i)}{\|f_{s,l}(e^i)\|_2} \right\|_1 . \tag{4}$$

$L_{d,t}'$ would be furhter added to update the distillation loss in Eq. 3. The final loss consists of the downstream task loss, $L_{task}$, *e.g.,* classification loss in VCR and the newly updated distillation loss with the distillation weight, $w$:

$$L_{final} = L_{task} + w \cdot L_d = L_{task} + w \cdot (E \cdot L_{d,v} + F \cdot (L_{d,t} + L_{d,t}')) . \tag{5}$$

### 3.3 Adaptive Two-Step Finetuning Strategy

Domain gap exists when directly transfering unimodal pre-trained knowledge to a specific downstream task. This may be due to the distribution differences between the pre-training and downstream data and discrepancy in optimization directions. With the aim to ease the gap, we design an Adaptive Two-step Finetuning.

**Adaptive Finetuning with Contrastive Learning:** We first finetune the base student VL model on a downstream dataset with Image-Text Matching (ITM). Moreover, we also incorporate the knowledge distillation on parallel as an additional task. As the base student model is adapting to the downstream dataset, the distilled knowledge from pre-trained teacher models can assist the student to learn intra-sample differentiation, differences between image-text pairs. Specifically, on VCR, we convert every question and correct answer pair into a text statement with heuristic rules (details in A.1.3) and regard each image-text pair as the positive sample. We also apply similar procedures on VQA and SNLI-VE. For each positive sample, we create two corresponding negative samples. In the first negative sample, we swap the image with another randomly selected one; In the second, we swap the text.

During finetuning, we extract the representative text features and visual features to represent the text and visual sequence respectively. These two features are then input into a prediction head, a MLP module, to predict the probability that the input image and text is a correct pair. The contrastive learning of ITM is via calculating the InfoNCE loss (Oord et al., 2018):

| Vision | Lang. | Conf. W. | T.T. Select. | T.S. Fine. | 100/Type | 1000/Type | Full |
|--------|-------|----------|--------------|------------|----------|-----------|------|
| | | | | | 30.85 | 53.48 | 75.53 |
| ✓ | | | | | 34.24 | 54.53 | 75.92 |
| | ✓ | | | | 35.96 | 55.04 | 76.15 |
| ✓ | ✓ | | | | 36.78 | 55.91 | 76.37 |
| ✓ | ✓ | ✓ | | | 37.21 | 57.64 | 76.83 |
| ✓ | ✓ | ✓ | ✓ | | 38.28 | 58.68 | 77.15 |
| ✓ | ✓ | ✓ | ✓ | ✓ | 40.43 | 58.98 | 77.61 |

Table 1: Ablation experiments using VL-BERT as the base student model and evaluated on the VCR standard validation set. The visual teacher is a pre-trained SwimTransformer V2 and the language teacher is a pre-trained RoBERTa. Vision represents vision distillation and Lang. represents language distillation. Conf. W. represents Distillation with Confidence Weighting. T.T. Select. represents Distillation with Text Token Selection. T.S.Fine. represents Adaptive Two-Step Finetuning.

$$L_{ITM} = -\log \frac{\exp\left(\text{sim}\left(z_i^V, z_i^T\right)/\tau\right)}{\exp\left(\text{sim}\left(z_i^V, z_i^T\right)/\tau\right) + \exp\left(\text{sim}\left(z_i^V, z_j^T\right)/\tau\right) + \exp\left(\text{sim}\left(z_k^V, z_i^T\right)/\tau\right)} \quad (6)$$

where $i \neq j$ and $i \neq k$. $sim$ represents the predicted probability by the prediction head. $\tau$ is a hyperparameter temperature (A.1.2). $z_i^V$ and $z_i^T$ are the visual and text representative features for a positive pair. $z_k^V$ and $z_j^T$ are the visual and text representative features of other samples.

$$\begin{cases} \text{1st stage,} & L_{final} = L_{ITM} + w \cdot L_d \\ \text{2nd stage,} & L_{final} = L_{task} + w \cdot L_d \end{cases} \quad (7)$$

**Final Finetuning:** Lastly, we finetune the base student model with the downstream task specific loss and also keep the knowledge distillation on parallel. The final loss is listed as above.

# 4 EXPERIMENT

## 4.1 MODEL

**Teacher Model:** As mentioned in Sec. 2, for a comprehensive comparison, we experiment with a wide range of encoders from unimodal pre-training or VL pre-training with image-text pairs. For the pre-trained visual encoders, we utilize ViT, SwinV2, CLIP-V and SLIP-V; For the text pre-trained encoders, we utilize RoBERTa, CLIP-T and SLIP-T.

**Student Model:** We adapt top-performing and publicly available VL models for highly-semantic VL tasks, including UNITER (Chen et al., 2020), VL-BERT (Su et al., 2020), and VILLA (Gan et al., 2020). All of these models represent variations of multimodal transformer-based architectures.

## 4.2 DATASET AND EVALUATION

To verify our claim that the unimodal pre-trained vision and language models can help to improve the permanence on downstream VL tasks. We evaluate ADVL across the three popular highly-semantic VL datasets: VCR, VQA and SNLI-VE under three settings: low-shot, full-data and domain-shifted settings.

**Low-shot:** Thre are 7 question types in VCR, 3 in SNLI-VE and 8 in VQA. For a better comparison, we design to have two standardized evaluation settings for all of them: (1) Selecting 100 images for each type, (100/Type) and (2) Selecting 1,000 images for each type, (1000/Type) (More details in Appendix A.3). Few-shot results are averaged over 4 runs (see Appendix A.4.1 A.4.2 for standard deviations).

**Domain-shift:** For VCR, we follow an existing domain-shifted setting (Ye & Kovashka, 2021) (Appendix A.3.1). For SNLI-VE and VQA, we also follow similar procedures in (Ye & Kovashka, 2021) to replace pronouns based on list of heuristic rules to create domain-shifted settings. (Appendix A.3.2, A.3.3)

| Teacher Model | | VCR | | | SNLI-VE | | | VQA | | |
|---|---|---|---|---|---|---|---|---|---|---|
| Vision | Lang. | 100/Type | 1000/Type | Full | 100/Type | 1000/Type | Full | 100/Type | 1000/Type | Full |
| - | - | 34.84 | 57.01 | 78.27 | 58.47 | 67.16 | 79.64 | 37.18 | 65.75 | 72.11 |
| SLIP-V | SLIP-T | 38.92 | 57.38 | 78.28 | 58.67 | 67.23 | 79.37 | 39.07 | 65.89 | 72.14 |
| CLIP-V | CLIP-T | **42.04** | 59.42 | 78.61 | 59.04 | 68.37 | 79.96 | 39.34 | 66.10 | 72.43 |
| CLIP-V | RoBERTa | 40.71 | 59.36 | 78.93 | 58.77 | 68.69 | 79.75 | 39.12 | 66.34 | 72.79 |
| ViT | RoBERTa | 39.05 | 58.14 | 78.42 | 58.44 | 67.56 | 79.24 | 39.01 | 66.17 | 72.49 |
| SwinV2 | RoBERTa | 41.87 | **60.18** | **79.13** | **59.07** | **68.83** | **80.87** | **39.67** | **66.96** | **73.04** |

Table 2: Knowledge distillation from combinations of paired encoders for VL downstream tasks. Vision represents vision teacher model and Lang. represents language teacher model.

### 4.3 ABLATION

As in Tab. 1, we performed an ablation study of ADVL with VL-BERT as the student model on VCR under the low-shot and full-data settings. The goal here is to understand how each ADVL component improves the performance. These results demonstrate 2 key observations: 1) Language distillation contributes more than vision distillation (as studied in previous works (Shen et al., 2021)). Hence it is incredibly important to include it in multimodal distillation. Further, distilling both vision and language perform better than either one alone. 2) Each component inside ADVL can bring consistent improvement over the course of evaluation on the validation set of VCR-Q2A.

**Distillation with Various Teachers:** Our method is flexible that we can have separate vision and text encoders as teacher models and conduct modality-specific distillation separately.

For demonstrating the benefits of distilling knowledge from unimodal pre-trained encoders for downstream VL tasks, we experiment ADVL on top of VILLA Gan et al. (2020) as the base model to distill from mutliple combinations of teacher models. Comparing among the first three rows, we observe that when applying ADVL to distill from dual encoders of CLIP and SLIP the downstream performance can be improved. Moreover, a vision or text encoder from a VL pre-training can also pair with a unimodal encoder pre-trained with image-only or text-only data to serve as the dual teachers. As in the 4th row of Tab. 2, ADVL even enables downstream VL models to flexibly distill learned knowledge of CLIP-V from VL pre-training and learned knowledge of RoBERTa from unimodal pre-training. Lastly, we also pair two independently unimodal pre-trained encoders as in the final two rows. We can spot that with Swin Transformer V2 and RoBERTa, we can achieve the best performance among the different combinations of paired dual-teachers. This further verifies the value of leveraging knowledge in unimodal pre-training to downstream VL tasks.

During experiments, we discover that combinations of teacher models correspond to different settings of picking attention layers within the student VL model as the optimal target distillation layers. When distilling from two independently unimodal pre-trained encoders, we find that selecting the center intermediate layer of the student VL model would deliver the best performance. For instance, select the 12th intermediate layer as the target distillation layer out of VL-BERT$_L$'s 24 layers. When distilling from vision and text encoders of a VL pre-training framework, such as CLIPs, selecting the final hidden layer of the student VL model is optimal. Refer to Appendix A.4.1 for more ablation.

### 4.4 COMPARISON WITH BENCHMARKS

#### 4.4.1 VCR

Results on the VCR dataset for Q2A, QA2R and Q2AR are shown in Tab. 3. For Q2A, we also conduct evaluations with several data availability and domain-shifted constraints. Based on our former evaluation with different dual-teachers, Tab. 2, we apply ADVL with the overall best dual-teacher combination: SwinV2 and RoBERTa. Although ADVL avoids redoing any pre-training, for comparison, we also include a recent top-performing method utilizing CLIP vision encoder with additional pre-training here for comparison (CLIP-ViL$_p$). We find ADVL can improve downstream VL models consistently across different settings, and surprisingly bringing stronger boost on low-scale and domain-shifted settings.

ADVL with VILLA delivers high performance on two public leaderboards: (1) On the AI2 hosted public leaderboad, it achieves a new SOTA across three metrics with (Q2A: **80.9%** QA2R: **83.3%**

| Model | Q2A | | | | | QA2R | | Q2AR | |
|---|---|---|---|---|---|---|---|---|---|
| | Val. | | | | Test | Val. | Test | Val. | Test |
| | 100/Type | 1000/Type | Full | Full/Domain-shifted | | | | | |
| VL-BERT | 30.85 | 53.48 | 75.53 | 71.13 | 75.8 | 77.91 | 78.4 | 58.85 | 59.7 |
| VL-BERT★ | 40.43 (+9.58) | 58.98 (+5.5) | 77.61(+2.08) | 74.55 (+3.42) | | 79.02 (+1.11) | | 61.33 (+2.48) | |
| UNITER | 31.43 | 54.24 | 76.67 | 73.84 | 77.3 | 79.98 | 80.8 | 61.32 | 62.8 |
| UNITER★ | 42.23 (+10.8) | 59.88 (+5.64) | 77.05 (+0.38) | 74.94 (1.1) | | 80.57 (+0.59) | | 62.08 (+0.76) | |
| VILLA | 34.84 | 57.01 | 78.27 | 75.43 | 78.9 | 82.33 | 82.8 | 64.44 | 65.7 |
| VILLA★ | 41.87 (+7.03) | 60.18 (+3.17) | 79.13 (+0.86) | 76.32 (+0.89) | 79.6 | 82.57 (+0.24) | 82.9 | 65.34 (+0.90) | 66.2 |
| CLIP-ViL$_p$ | 34.63 | 53.54 | 68.36 | 66.83 | - | - | - | - | - |

Table 3: Results on VCR with different VL architectures. A model with ★ represents we apply ADVL on top of it. Baseline results are based on our re-implementation of the VL models. Val. represents evaluation on validation sets.

| Model | SNLI-VE | | | | | VQA | | | | | |
|---|---|---|---|---|---|---|---|---|---|---|---|
| | Val. | | | | Test | Val. | | | | Test-Val | Test-Std |
| | 100/Type | 1000/Type | Full | Full/Domain-shifted | | 100/Type | 1000/Type | Full | Full/Domain-shifted | | |
| VL-BERT | 53.28 | 62.31 | 74.66 | 70.96 | 74.02 | 35.33 | 63.29 | 69.05 | 67.68 | 71.79 | 72.22 |
| VL-BERT★ | 56.78 | 65.37 | 75.75 (+1.09) | 73.19 (+2.23) | 75.43 | 37.12 | 65.71 | 71.42 (+2.37) | 70.55 (+2.87) | | |
| UNITER | 58.36 | 66.23 | 79.02 | 76.72 | 79.19 | 36.46 | 64.43 | 71.26 | 69.88 | 73.82 | 74.02 |
| UNITER★ | 59.42 | 68.34 | 80.14 (+1.12) | 78.54 (+1.82) | 80.23 | 39.75 | 65.93 | 71.94 (+0.68) | 70.18 (+0.3) | | |
| VILLA | 58.47 | 67.16 | 79.64 | 77.22 | 79.32 | 37.18 | 65.75 | 72.11 | 70.18 | 74.69 | 74.87 |
| VILLA★ | 59.96 | 68.83 | 80.87 (+1.23) | 79.56 (+2.34) | 80.32 | 39.67 | 66.96 | 73.04 (+0.93) | 72.74 (+2.56) | 75.81 | 76.04 |
| CLIP-ViL$_p$ | 59.48 | 68.32 | 80.61 | 78.58 | 80.2 | 39.01 | 66.84 | 73.91 | 71.46 | 76.48 | 76.70 |

Table 4: Results on SNLI-VE and VQA. A model with ★ represents we apply ADVL on top of it. Baseline results are based on our re-implementation of the VL models.

Q2AR: **67.7%**) compared to other published single models pre-trained with image-text data (2) On UW public leaderboard: With (Q2A: **79.6%** QA2R: **82.9%** Q2AR: **66.2%**), it also achieves a new SOTA of Q2A performance compared to other published single models pre-trained with image-text data, as well as overall competitive performance for QA2R and Q2AR.

### 4.4.2  SNLI-VE AND VQA

Results on SNLI-VE and VQA are shown in Tab. 4. We observe that ADVL improves downstream VL models consistently across different settings on both SNLI-VE and VQA. The improvement on low-scale and domain-shifted settings is even more significant. Besides, on SNLI-VE, ADVL outperforms concurrent work CLIP-ViL$_p$ under all evaluated conditions. On VQA, ADVL outperforms finetuning approach CLIP-ViL$_p$ under the few-shot settings and achieves comparable performance to CLIP-VIL$_p$ under the full-shot setting. Furthermore, under the domain-shifted setting, ADVL outperforms CLIP-VIL$_p$ by a large margin. Applying ADVL on top of an ensemble of VL models, such as VILLA and VL-BERT, lead to **74.29 %** on local validation, outperforming CLIP-ViL$_p$.

### 4.5  INCREASED UTILIZATION OF VISION MODALITY

ADVL improves VL models on varioius downstream tasks. After carefully examining why ADVL improves VL models, we are surprised to discover that existing VL models underutilized vision information. In particular, following Cao et al. (2020), we measure Modality Importance (MI) of vision and language modalities and plot the result in Fig 8 (see appendix). We observe that ADVL increases the Vision MI as shown in Fig 8 (a) and the MI difference between Vision and Text is decreased as shown in Fig 8 (b). Both observations indicate that ADVL encourages base VL models to better utilize vision information. Details are in the Appendix.

## 5  CONCLUSIONS

We initiated this study with the motivation to explore how to utilize the unimodal pre-trained knowledge to help the finetuning in downstream VL tasks. Instead of fusing VL information in the pre-training stage, we focus on the finetuning stage only. With our new approach, ADVL, we can flexibly distill pre-trained knowledge from different pairs of vision and text encoders. With extensive experiments, we verify that unimodal pre-trained knowledge is beneficial to direct finetuning on downstream VL tasks.

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

## A APPENDIX

### A.1 TRAINING/IMPLEMENTATION DETAILS

#### A.1.1 ADAPTIVE TEXT TOKEN SELECTION

With experiments, we find that when the number of selected tokens is 2 ($m = 2$), Adaptive Text Token Selection would produce the highest performance.

#### A.1.2 ADAPTIVE TWO-STEP FINETUNING

$\tau$ is a hyperparameter temperature and set to 1 in all experiments.

#### A.1.3 HEURISTIC RULES FOR CONVERSION

When conducting Adaptive Finetuning with Contrastive Learning on VCR, we convert every pair of question and correct answer choice into a text statement with heuristic rules as in Fig. 4. When we conduct Finetuning with Contrastive Learning on VQA, we directly concatenate the question and the correct answer together as the text statement. When we conduct Finetuning with Contrastive Learning on SNLI-VE, we only keep the images with entailed hypothesises as positive pairs. The hypothesises are already in statement format so we do not need to further convert them.

### A.2 EXPERIMENT SET-UP

All the baseline experiment results with models including VL-BERT, UNITER, VILLA and CLIP-ViL$_p$ are based on code provided by the authors, which we modified to include our distillation methods. CLIP-ViL$_p$ did not originally evaluate on VCR in their pre-print. However, we evaluated it on the VCR dataset.

```
question = "What is going to happen next?"
answer = "Person2 is going to say how cute the childn are."

question = question.spit(" ")
answer = answer.split(" ")

if 'what' == question[0]:
    if 'will' == question[1] and 'happen' == question[2] and 'next' == question[3] and '?' == question[4]:
        statement = answer
        template_type = '1a'
    elif 'if' in question:
        statement = question[question.index('if'):-1] + answer
        template_type = '1b'
    elif 'after' in question:
        statement = question[question.index('after'):-1] + answer
        template_type = '1c'
    elif 'is' == question[1] or 'are' == question[1] or 'was' == question[1] or 'has' == question[1] or 'do' == question[1] or 'did' == question[1]\
            or 'does' == question[1] or 'were' == question[1]:
        statement = answer
        template_type = '1d'
    elif 'would' == question[1]:
        statement = answer
        template_type = '1e'
    elif 'will' == question[1] or 'might' == question[1]:
        statement = answer
        template_type = '1f'
elif 'whose' == question[0]:
    if 'is' in question:
        statement = answer + question[question.index('is'):-1]
        template_type = '2a'
    elif 'are' in question:
        statement = answer + question[question.index('are'):-1]
        template_type = '2a'
elif 'why' == question[0]:
    if 'does' == question[1] or 'do' == question[1] or 'are' == question[1] or 'is' == question[1] or 'did' == question[1] or 'would' == question[1]\
            or 'was'==question[1] or 'has' == question[1]:
        statement = [question[2]] + [question[1]] + question[3:-1] + ['because'] + answer
        template_type = '3a'
elif 'how' == question[0]:
    if 'if' in question:
        statement = question[question.index('if'):-1] + answer
        template_type = '4a'
    elif ('feel' in question) or ('feeling' in question):
        statement = answer
        template_type = '4b'
    elif 'do' == question[1] or 'will' == question[1] or 'is' == question[1]:
        statement = answer
        template_type = '4c'
elif 'where' == question[0]:
    statement = answer
    template_type = '5a'
elif ('is' == question[0]) or ('can' == question[0]) or ('does' == question[0]) or ('do' == question[0]) or ('was' == question[0]) \
        or 'are' == question[0] or 'did' == question[0] or 'will' == question[0] or 'doesn' == question[0] or 'would'==question[0]:
    if 'yes' in answer or ('yes' not in answer and 'no' not in answer):
        statement = [question[1]] + [question[0]] + question[2:-1] + answer[1:]
    else:
        statement = [question[1]] + [question[0]] + ["not"] + question[2:-1] + answer[1:]
    template_type = '6a'

elif 'who' == question[0]:
    statement = answer
    template_type = '7a'
elif 'which' == question[0]:
    statement = answer
    template_type = '8a'
```

Figure 4: Heuristic rules for converting question and answer pairs into statements. The input question and answer by default are in format of string.

### A.2.1 TEACHER MODEL

For ViT, we utilize the ViT$-L_{32}$ version pre-trained on ImageNet22k.

For Swin Transformer V2, we utilize the SwinV2$-L$ version pre-trained on ImageNet22k.

For RoBERTa, we utilize the pre-trained RoBERTa$-L$ without further finetuning.

For CLIP, we utilize the ViT$-B_{32}$ version pre-trained with 400 Million paired image-text data.

### A.2.2 STUDENT MODEL

**- VL-BERT:** We train our model for 30 epochs with warm-up steps of 1000, SGD optimizer. Initial learning rate is $7.0e-5$ and decays by 0.1 at the 14th, 18th and 26th epoch. The gradient

accumulation steps is set to be 4 on 8 NVIDIA V100 GPUs (32GB VRAM). The total number of attention layers for VL-BERT$_{Large}$ is 24.

**- UNITER:** Started with warm up steps of 800, the model is trained with total steps of 8000. With AdamW optimizer, the intial learning rate is set to be $6e - 05$ with weight decay of 0.01 and batch size of 4000. The gradient accumulation steps is set to be 5 on 4 NVIDIA TITAN RTX GPUs (24GB VRAM). The total number of attention layers for UNITER$_{Large}$ is 24.

**- VILLA:** Warm up steps is set to be 1000 and total training steps is 10000. The intial learning rate is $6e - 05$ with weight decay of 0.01 and AdamW optimizer. The training batch size is 1250. The gradient accumulation steps is set to be 8 on 8 NVIDIA TITAN RTX GPUs (24GB VRAM). The total number of attention layers for VILLA is 24.

### A.2.3 COMPARISON MODEL

**- CLIP-ViL$_p$:** The model is trained for 20 epochs with batch size of 24. The optimizer is AdamW with a peak learning rate of 5 x $10^{-5}$.

### A.3 EVALUATION SETTING

### A.3.1 VISUAL COMMONSENSE REASONING (VCR)

The VCR benchmark presents images along with a paired question, a set of candidate answers, and a set of candidate rationales (Zellers et al., 2019). The dataset includes 290k questions, in reference to 110k unique visual scenes. The questions are constituted into 7 categories based on patterns in the questions. Please see the supplementary material for a full list.

**VCR Dataset Question Sub-Types**   According to Zellers et al. (2019), among VCR questions, 38% fall into explanation (why, how come, *etc.*), 24% activity (doing, looking, event, *etc.*), 13% temporal (happened, before, after, *etc.*), 8% mental (feeling, thinking, *etc.*), 7% role (relations, occupations, *etc.*), 5% scene (where, near, *etc.*), and 5% hypothetical (if, would, could, *etc.*). Details can be referred to Zellers et al. (2019).

**Low-Shot:**   In the low-shot setting, we have 2 training set partitions of varying sizes. Since VCR has **7 types** of questions thus (1) we select 100 samples **per question category**, (1000/Type), totalling 700 pairs, or 0.3% of the entire dataset, and (2) 1,000 samples per category, (1000/Type), totalling 7,000 pairs, or 3%. Each experiment is run 4 times.

**Domain-shifted:**   We refer to an existing domain-shifted evaluation configuration Ye & Kovashka (2021) that focuses on changing pronouns between question and answers.

In questions and answer choices of VCR, a person can be referred by a person tag ([person2] or [2]) or a pronoun (he, she). The method applies simple rule-based modification on the person tags and pronouns in VCR answer choices.

For the **correct answer choice,** the rule is to remove the person tag and swap it with a pronoun, instead.

For the **incorrect answer choice,** the rule is to make the option more associated to the question by repeating the person that is asked, as in Fig. 7.

Some examples are shown in Fig. 5 and Fig. 6

### A.3.2 VISUAL ENTAILMENT (SNLI-VE)

The Stanford Natural Language Inference Visual Entailment (SNLI-VE) task Xie et al. (2019) presents images as a premise, with a paired hypothesis text. The goal is to predict whether the image entails or contradicts the hypothesis, or whether neither is the case (neutral).

**Low-shot:**   There are only 3 classes of answer labels corresponding to 3 types of relationship between the image premise and the text hypothesis (entailment, neutral, and contradiction). Different

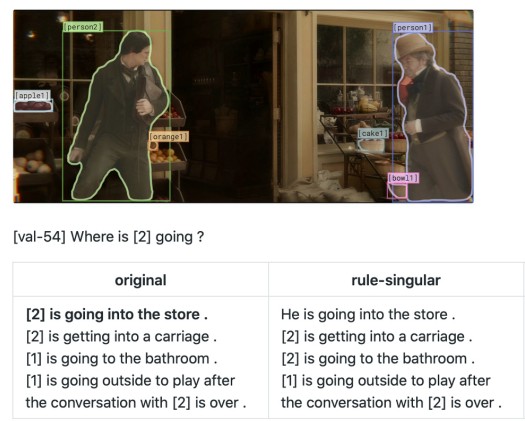

[val-54] Where is [2] going ?

| original | rule-singular |
|---|---|
| **[2] is going into the store .** | He is going into the store . |
| [2] is getting into a carriage . | [2] is getting into a carriage . |
| [1] is going to the bathroom . | [2] is going to the bathroom . |
| [1] is going outside to play after the conversation with [2] is over . | [1] is going outside to play after the conversation with [2] is over . |

Figure 5: Rule-based modification for one person.

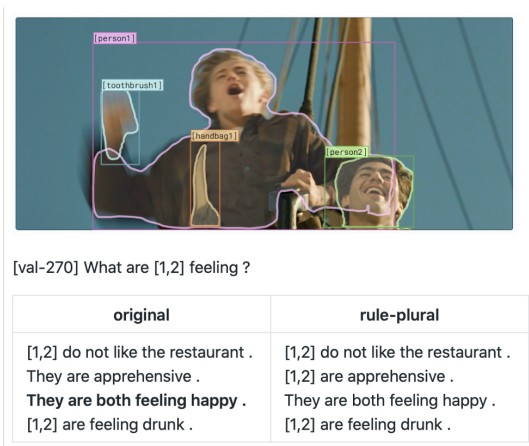

[val-270] What are [1,2] feeling ?

| original | rule-plural |
|---|---|
| [1,2] do not like the restaurant . | [1,2] do not like the restaurant . |
| They are apprehensive . | [1,2] are apprehensive . |
| **They are both feeling happy .** | They are both feeling happy . |
| [1,2] are feeling drunk . | [1,2] are feeling drunk . |

Figure 6: Rule-based modification for multiple people.

Examples - Modifying the distracting options.

| Question | Original | Changed to |
|---|---|---|
| *Why is [person2] in such a rush?* | *He used the wrong ingredients to make the meal.* | *[person2] used the wrong ingredients to make the meal.* |
| *How is [person2] feeling?* | *[person1] is very excited.* | *[person2] is very excited.* |

Figure 7: Examples of modifying the incorrect answer choices.

from VCR, an image can pair with around 5 text premises in SNLI-VE. Therefore, we choose the image-based selection. We also have two settings: (1) We select 100 random images for each class, *300* images in total paired with around *1,500* text premises. (2) 1,000 randomly sampled for each class label, then *30,00* samples in total with around *15,000* premises. They both correspond to $0.3\%$ and $3\%$ of SNLI-VE. Each experiment is run 4 times.

**Domain-shifted:** We follow similar procedures as in Ye & Kovashka (2021) to only replace pronouns based on list of heuristic rules to create domain-shifted settings, as in Tab. 5

| Original Words | Replaced Words |
|:---:|:---:|
| Man | Male |
| Boy | |
| Woman | Female |
| Girl | |
| People | Persons |

Table 5: Heuristic rules of creating domain-shifted setting on SNLI-VE.

| Types | Original Words | Replaced Words |
|:---:|:---:|:---:|
| Pronouns | Man | Person |
| | Woman | |
| | He | |
| | She | |
| Question Types | How many xxx? | What is the number of xxx? |
| | What is xxx doing? | What is the action of xxx? |
| | Who xxx? | Which person xxx? |
| | What color xxx? | Which color xxx? |

Table 6: Heuristic rules of creating domain-shifted setting on VQA.

### A.3.3 VISUAL QUESTION ANSWERING (VQA)

Different from VCR and VE, for every image-question pair in VQA Antol et al. (2015), question-specific multiple choices are not provided. Instead, the global set of all possible answer choices for all the questions are provided (more than 3,000). The challenge is then to select the correct answer choice from this set for the given image-question pair.

**Low-shot:** There is not a clear categorization of VQA questions. Based on our analysis of the first n-gram words of questions, we group and finalize to 8 types in total. In VQA, an image is also paired with several questions, 5.4 on average. Thus, we also rely on image-based sampling. We have two settings of image sampling in low-shot settings: 100 random images per question type and 1,000 random image per question type. After collecting up to 2 paired questions per each selected image, we have two low-shot set: a set with around 1,600 questions and another set with 16,000 questions. They both correspond to $0.3\%$ and $3\%$ of VQA. Each experiment is run 4 times.

**Domain-shifted Setting:** We follow similar procedures as in Ye & Kovashka (2021) to replace pronouns and initial words of questions based on list of heuristic rules to create domain-shifted settings, as in Tab. 6

### A.4 EVALUATION RESULT

### A.4.1 ABLATION EVALUATION

**Different Teacher Models**

**Distillation with Different Layers** We conduct ablation study with ADVL on top of VILLA as the student model. The ablation study is only evaluated on VCR. As in Tab. 9, we can observe that: (1) when the pre-trained image and text encoders are from a VL pre-training, the optimized target attention layer should be set at the last hidden layer of the VL student model. (2) when the pre-trained image and text encoders are from a unimodal pre-training, the optimized target attention layer should be set at an intermediate layer of the VL student model. Typically, this would be one half of the VL student model's total number of attention layers.

### A.4.2 LOW-SHOT EVALUATION

Refer to Tab. 7, Tab. 8, Tab. 9, Tab. and 10.

| Vision | Lang. | Conf. W. | T.T. Select. | T.S. Fine. | 100/Type | | 1000/Type | |
|:---:|:---:|:---:|:---:|:---:|:---:|:---:|:---:|:---:|
| | | | | | Accuracy | Var | Accuracy | Var |
| | | | | | 30.85 | 1.12 | 53.48 | 0.74 |
| ✓ | | | | | 34.24 | 0.93 | 54.53 | 0.54 |
| | ✓ | | | | 35.96 | 0.59 | 55.04 | 0.70 |
| ✓ | ✓ | | | | 36.78 | 0.91 | 55.91 | 0.82 |
| ✓ | ✓ | ✓ | | | 37.21 | 1.40 | 57.64 | 1.03 |
| ✓ | ✓ | ✓ | ✓ | | 38.28 | 1.04 | 58.68 | 0.68 |
| ✓ | ✓ | ✓ | ✓ | ✓ | 40.43 | 1.36 | 58.98 | 1.09 |

Table 7: Ablation experiments using VL-BERT as the base student model and evaluated on the VCR standard validation set. The visual teacher is a pre-trained SwimTransformer V2 and the language teacher is a pre-trained RoBERTa. Vision represents vision distillation and Lang. represents language distillation. Conf. W. represents Distillation with Confidence Weighting. T.T. Select. represents Distillation with Text Token Selection. T.A.Fine. represents Adaptive Two-step Finetuning.

| Teacher Model | | VCR | | | | SNLI-VE | | | | VQA | | | |
|:---:|:---:|:---:|:---:|:---:|:---:|:---:|:---:|:---:|:---:|:---:|:---:|:---:|:---:|
| Vision | Lang. | 100/Type | | 1000/Type | | 100/Type | | 1000/Type | | 100/Type | | 1000/Type | |
| | | Accuracy | Var | Accuracy | Var | Accuracy | Var | Accuracy | Var | Accuracy | Var | Accuracy | Var |
| - | - | 34.84 | 0.83 | 57.01 | 0.69 | 58.47 | 1.32 | 67.16 | 1.51 | 37.18 | 0.90 | 65.75 | 1.23 |
| SLIP-V | SLIP-T | 38.92 | 0.54 | 57.38 | 0.38 | 58.67 | 0.95 | 67.23 | 1.19 | 39.07 | 0.85 | 65.89 | 0.84 |
| CLIP-V | CLIP-T | **42.04** | 0.88 | 59.42 | 1.06 | 59.04 | 1.47 | 68.37 | 1.07 | 39.34 | 0.75 | 66.10 | 1.41 |
| CLIP-V | RoBERTa | 40.71 | 1.26 | 59.36 | 1.05 | 58.77 | 1.85 | 68.69 | 1.44 | 39.12 | 1.13 | 66.34 | 0.77 |
| ViT | RoBERTa | 39.05 | 1.01 | 58.14 | 0.92 | 58.44 | 0.86 | 67.56 | 0.75 | 39.01 | 0.52 | 66.17 | 0.54 |
| SwinV2 | RoBERTa | 41.87 | 0.93 | **60.18** | 0.78 | **59.07** | 0.70 | **68.83** | 0.89 | **39.67** | 0.99 | **66.96** | 1.06 |

Table 8: Knowledge distillation from combinations of paired encoders for VL downstream tasks. Vision represents vision teacher model and Lang. represents language teacher model.

### A.4.3 ENSEMBLE RESULTS

**VCR:** As our approach is model agnostic, we further apply ADVL on top of multiple ensembled VL base models (*i.e.,* VL-BERT and VILLA) and achieve further significant gains in performance (Q2A: **80.93%**, QA2R: **84.01%**) in the fully-sampled validation set.

**VQA:** Further experiments with applying ADVL on top of an ensemble of VL models, *i.e.,* VILLA and VL-BERT can even achieve **74.29 %** on local validation, outperforming CLIP-ViL$_p$.

### A.5 ANALYSIS

Based on our results, we have seen the impact on end task performance of distilling knowledge from largely pre-trained unimodal encoders into student VL models. However, a key question still remains: how much the improvement comes from the distillation of each modality respectively?

Following Cao et al. (2020), we measure the Modality Importance (MI) of both visual modality and textual modality. This approach sums the attention weights across heads of each modality to understand how much each modality is weighted by the model. Fig. 8 shows the average the MI values of all the heads for each layer on VL-BERT, both with and without MD, trained on the VCR dataset. One can clearly observe that prior to distillation, the model more heavily weights the text modality as being important to correctly choosing answers. After distillation, both vision and text modalities are more equally considered. This may also explain why the model yields such impressive performance improvements in low-shot and domain shifted scenarios.

Fig. 8, we plot the MI values of all the heads across 12 layers in VL-BERT Base and VL-BERT Base with MD. It is obvious that, at the last layer, the textual MI heatmap on the right is denser than the visual MI heatmap on the left. This shows a common flaw from existing V+L models that they heavily rely on the textual information than the visual part indicating the shallow understanding of the visual scene in downstream tasks. However, in the bottom row, the difference between the left and right heatmaps is much smaller and the visual MI heatmap at the bottom is also clearly more denser than the one at the top.

| Teacher Model | | Target Attention Layer / Total of Attention Layers | VCR | | |
|---|---|---|---|---|---|
| Vision | Lang. | | 100/Type | 1000/Type | Full |
| - | - | - | 34.84 | 57.01 | 78.27 |
| CLIP-V | CLIP-T | 24th / 24 | **42.04** | 59.42 | **78.61** |
| | | 20th / 24 | 40.06 | 59.73 | 78.03 |
| | | 16th / 24 | 38.53 | **58.60** | 78.46 |
| | | 12th / 24 | 37.70 | 58.96 | 78.00 |
| SwinV2 | RoBERTa | 24th / 24 | 26.74 | 38.99 | 78.03 |
| | | 20th / 24 | 28.81 | 45.39 | 78.28 |
| | | 16th / 24 | 35.93 | 53.10 | 78.67 |
| | | 12th / 24 | **41.87** | **60.18** | **79.13** |

Table 9: Knowledge distillation on different target layers of VL models for VCR. Vision represents vision teacher model and Lang. represents language teacher model.

| Model | SNLI-VE | | | | VQA | | | |
|---|---|---|---|---|---|---|---|---|
| | 100/Type | | 1000/Type | | 1000/Type | | 1000/Type | |
| | Accuracy | Var | Accuracy | Var | Accuracy | Var | Accuracy | Var |
| VL-BERT | 53.28 | 1.04 | 62.31 | 1.28 | 35.33 | 0.75 | 63.29 | 0.69 |
| VL-BERT⋆ | **56.78** | 1.48 | **65.37** | 0.51 | **37.12** | 0.85 | **65.71** | 0.49 |
| UNITER | 58.36 | 0.87 | 66.23 | 0.85 | 36.46 | 0.24 | 64.43 | 0.48 |
| UNITER⋆ | **59.42** | 1.14 | **68.34** | 1.32 | **39.75** | 0.51 | **65.93** | 1.07 |
| VILLA | 58.47 | 1.32 | 67.16 | 1.51 | 37.18 | 0.90 | 65.75 | 1.23 |
| VILLA⋆ | **59.96** | 0.70 | **68.83** | 0.89 | **39.67** | 0.99 | **66.96** | 1.06 |
| CLIP-ViL$_p$ | 59.48 | 0.89 | 68.32 | 0.63 | 39.01 | 0.13 | 66.84 | 0.81 |

Table 10: Low-shot evaluation on SNLI-VE and VQA. A model with ⋆ represents we apply ADVL on top of it. Baseline results are based on our re-implementation of the VL models. Few-shot results are averaged over 4 runs. 1000/Type and 1000/Type represent two low-shot settings of 100 and 1000 samples per question type.

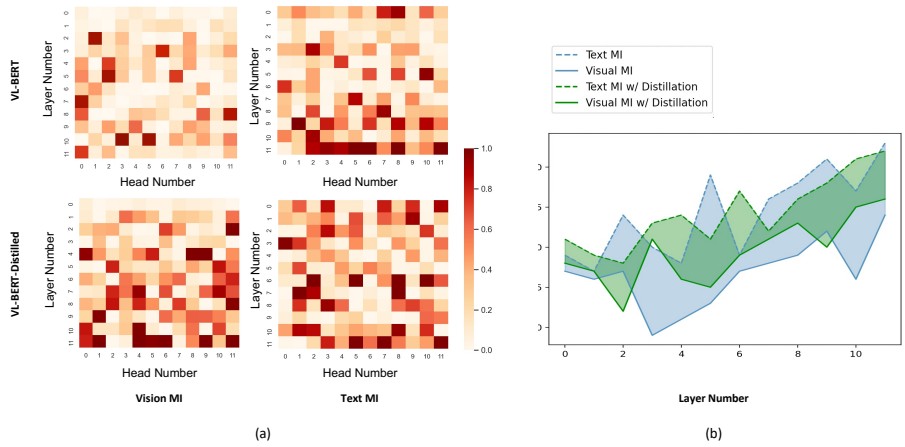

Figure 8: (a) Heatmap of MI values for both vision and text modality. Each cell represents a combination between 12 heads and 12 layers in VL-BERT$_B$ model. The top row corresponds to the original VL-BERT$_B$ model and the bottom row corresponds to VL-BERT$_B$ with ADVL. Index starts with 0. (b) Average Modality Importance (MI) values for each layer of VL-BERT$_B$ with ADVL (green) and baseline (blue). Shaded areas represent differences between vision and text modalities.

| Model | Q2A | | | |
| | 100/Type | | 1000/Type | |
| | Accuracy | Var | Accuracy | Var |
|---|---|---|---|---|
| VL-BERT | 30.85 | 1.12 | 53.48 | 0.74 |
| VL-BERT⋆ | **40.43 (+9.58)** | 1.36 | **58.98 (+5.5)** | 1.09 |
| UNITER | 31.43 | 1.32 | 54.24 | 0.49 |
| UNITER⋆ | **42.23 (+10.8)** | 0.76 | **59.88 (+5.64)** | 0.56 |
| VILLA | 34.84 | 0.83 | 57.01 | 0.69 |
| VILLA⋆ | **41.87 (+7.03)** | 0.93 | **60.18 (+3.17)** | 0.78 |
| CLIP-ViL$_p$ | 34.63 | 1.06 | 53.54 | 0.80 |

Table 11: Low-shot evaluation on VCR. A model with ⋆ represents we apply ADVL on top of it. Baseline results are based on our re-implementation of the VL models. Few-shot results are averaged over 4 runs. 1000/Type and 1000/Type represent two low-shot settings of 100 and 1000 samples per question type.

