# OpenReview forum: "ADVL: Adaptive Distillation for Vision-Language Tasks"
_ICLR.cc/2023/Conference — Submitted to ICLR 2023_

### Official Review · Reviewer_LFJd · 2022-10-25

**Confidence:** 5
**Correctness:** 3
**Technical Novelty And Significance:** 2
**Empirical Novelty And Significance:** 3
**Recommendation:** 3

**Clarity, Quality, Novelty And Reproducibility:**

The paper is clear and novel overall. It maybe easy to reproduce if the code will become available eventually.

**Strength And Weaknesses:**

Pros:
1. Experiments are conducted on many pretrained variants and datasets, and clearly show gains over VL baselines. This shows that the proposed method is effective.
2. Ablation studies are performed and it is clear that each proposed component is beneficial.
3. The paper is clear and easy to follow.

Cons:
1. The resulting student model is a large-scale VL pretrained checkpoint itself. It will be more interesting if the method can get strong performance on non-pretrained or smaller-scale models. As of now, it is hard to say the method brings 'efficiency' given everything is pretrained anyway.
2. The method contains too many specialized components. Although ablation does show the benefit of each, this still makes the method hard to use in practice. In fact, I am a bit hesitant to call it a 'method' and feel it is more like a bag of 'methods'.
3. Although some analysis is given, it is still hard to understand why the method is better. Since it is not intuitive why such framework can/should improve visual attention, it seems a must to dig further to understand details such as which exact step brings this benefit and why.
4. The proposed method is also designed for fusion-based VL models and not directly applicable for other trending architectures such as encoder-decoder.

**Summary Of The Paper:**

This paper studies how to utilize unimodal pretrained models to improve vision-language downstream tasks through distillation. In particular, the paper proposes to use both visual and textual teachers to supervise a student VL model during the finetuning stage. A group of methods/tricks have been explored, including adaptive confidence reweighting, adaptive token selections, and two-stage finetuning. The resultant framework improves the downstream VL performance on the student model.

**Summary Of The Review:**

In general, this paper presents a seemingly working framework to improve VL tasks with distillations from both vision and language pretrained checkpoints. However, the method is quite complicated and appears to be a bag of tricks. On the other hand, there is no clear and comprehensive understanding of the source of improvement. Therefore, I wish to see more studies to understand the trade-off between the additional training costs and true benefits the method brought.

---

### Official Review · Reviewer_JhCz · 2022-10-31

**Confidence:** 4
**Correctness:** 3
**Technical Novelty And Significance:** 3
**Empirical Novelty And Significance:** 2
**Recommendation:** 3

**Clarity, Quality, Novelty And Reproducibility:**

- The organization of the manuscript is good to read and easy to follow. However, so many typos included.
- Related work is well-surveyed.
- I think that the ideas on this paper are novel.
- The explanation of the methods is explicit and concrete. I think it is enough to be reproducible.


**Strength And Weaknesses:**

*Strength
- The proposed ideas are good to achieve the research goal. I think the proposed methods would be practically useful if they are generally applicable and the benefit is clear in most cases.
- The paper presents interesting findings on the performance improvements in the low-shot area.
- Various settings of experiments are performed and the results are consistent as the authors’ claims.


*Weaknesses
- I'm concerned with that the significance of this work would be limited. This method is not for building compact student models (in normal scenarios of knowledge distillation), but for improving the performance based on VL student models, unimodal teacher models for vision, and unimodal teacher models for language. The generality of the performance improvement is not clearly validated yet, and the low-shot setting, the area that the benefit is significant, needs the criteria of how low is appropriate to apply this method.
- One of the main claims is that the proposed methods achieve new SOTA on the VCR tasks. However, there are some problems on that paragraph. I checked the two leaderboards (1: https://leaderboard.allenai.org/vcr/submissions/public, 2: https://visualcommonsense.com/leaderboard/ ). Following the AI2 leaderboard, I realized that the numbers from the page 8 of the manuscript, “Q2A: 80.9%, QA2R:83.3%, Q2AR: 67.7%”, is the 1st one’s and it is for MERLOT, not for ADVL. The ADVL’s is “80.4%, 82.3%, 66.2%”. Following the VCR leaderboard, ADVL is the 18th runner and the score is “79.6, 82.9, 66.2”. Considering the submission dates of the others and the ranking, I cannot accept the claim currently.


**Summary Of The Paper:**

This paper proposes methods to distill unimodal teacher models of vision and language onto vision-language (VL) student models for utilizing the benefit of unimodal pre-trained models.
The authors present ideas to improve the performance: (1) VL student models are not changed, but additional learnable components are attached on the top of them, (2) instance-level weighting and (3) text token selection to consider for distillation.
The authors claim that the methods have good advantages to improve the performance and to avoid pre-training large models, and achieve new state-of-the-art performance.
The two former claims are validated with some experimental results on 3 kinds of vision-linguistic tasks (Vision Commonsense Reasoning, Visual QA, SNLI-SE) including ablation study and comparative analysis.
However, the latter one is NOT justified since the supporting evidences in the web-based leaderboards have some problems.

**Summary Of The Review:**

I think that the weaknesses on the significance of this work is so critical, and it is difficult to accept the authors' claims.

---

### Official Review · Reviewer_chYB · 2022-11-04

**Confidence:** 4
**Correctness:** 2
**Technical Novelty And Significance:** 1
**Empirical Novelty And Significance:** Not applicable
**Recommendation:** 3

**Clarity, Quality, Novelty And Reproducibility:**

- Not always clarified, especially experiment setups.
- Overall quality is not satisfactory.
- Novelty is limited.
- Reproducibility is expected as open-sourcing is promised. In addition, the method itself is straightforward to implement.

**Strength And Weaknesses:**

**Strengths**:
1. The motivation is clear. It is naturally expected that vision-language models can benefit from stronger unimodal encoders, for which the best strategies are not very well explored by the community.
2. The idea is simple and easy to implement. According to the evaluation, the proposed method shows its effectiveness especially when the finetuning data is limited.
3. Although the presentation can be polished, the overall narrative and explanation is clear and easy to follow.

**Weakness**
1. **Flawed experiment setup**. The motivation is to leverage unimodal data, which are assumed easier to obtain than image-text pairs. However, the teacher networks used in the experiments are not always uni-modal. For example, CLIP, SLIP are trained on a large amount of image-text pairs, meaning that their unimodal encoders are also aligned across modals. This contradicts with the motivation.
2. **Limited technical novelty and incremental empirical gains**. The proposed distillation loss is standard and by itself is not technically new. In addition, improvements on full-shot cases are mostly marginal. Considering this is the combined benefit of multiple techniques, e.g. distillation, text token selection, contrastive learning, I am not fully convinced by the empirical value of the proposed method.
3. **Student networks** are too weak to prove the proposed techniques are useful for the more recent (and more powerful) models. The selected student networks, VL-BERT, UNITER, VILLA, while they are great and highly reputable works in the community, their performance is not as competitive as for today. Therefore, the obtained task performance is far from state-of-the-art. For example, on VQA, more recent works (BEiT-3) achieve 84+, while the best reported result in the manuscript is ~76. To make the method more convincing, authors may consider use more recent VL models as student works, and try to further push their limits. For example, to use LLM such as GPT to improve the text representation of BEiT.
5. **More recent VL-pretrained works are not well acknowledged**. Although I understand the experiment setup, missing reference to more recent VL works prevent readers from getting a good research landscape in the multimodal pre-training. For example, recent great VL works, ALIGN, ALBEF, OFA, Frozen, Flamingo, Florence. BLIP, BEiT3. Authors are suggested to better position their work among the more recent ones.
4. It is not fully clarified what is the difference between the so-called "ITM" (image-text matching) and the contrastive losses used in other  VL pretrained models, such as ALIGN, ALBEF. Conventionally the ITM loss is a binary prediction task, while the particular one used in this work is more often referred as contrastive learning loss. This may create unnecessary confusions.
6. Presentation can be improved. Typos are not uncommon. Experiment setups are not entirely clear. For example, what is the experiment environment and training receipts.

**Summary Of The Paper:**

This work is concerned with vision-language representation learning. Main contribution is a new distillation technique that uses stronger uni-modal encoders as teacher network, to improve the finetuning performance of pre-trained VL models.

In specific, the work proposes Adaptive Distillation for Vision-Language (ADVL) tasks. ADVL leverages unimodal pre-trained models (e.g. ViT for vision, Roberta for text). The distillation is achieved by minimizing the L1 distance between features from the uni-modal teacher network, and the ones from the pre-trained VL student models. Authors also propose additional mechanisms, such as (i) adaptive confidence-based weighting, which adjust distillation weights based on student networks' confidence scores; (ii) adaptive text token selection, which enables the distillation process to focus on semantically more important text tokens; (iii) finetuning with contrastive learning to enhance cross-modal alignment.

Experiments are conducted using UNITER, VL-BERT, VILLA as student networks, and various unimodal encoders as teacher networks, including SLIP, CLIP ViT, Swin transformer. The proposed technique improves student networks on visual commonsense reasoning (VCR), visual entailment (SNLI-VE), and visual question answering (VQA) significantly in low-shot setups, and marginally in full-shot setups.

**Summary Of The Review:**

The proposed technique provides marginal empirical benefit while the method itself is technically incremental. Experiments are flawed, especially the choice of teacher networks falsifies the motivation of relying on uni-modal data only. It is also not clear whether the proposed technique will push the limits of more recent VL pre-trained models, especially those trained end-to-end. Therefore, I'd not recommend a rejection.

---

### Decision · Program_Chairs · 2023-01-20

**Decision:**

Reject

**Justification For Why Not Higher Score:**

N/A

**Justification For Why Not Lower Score:**

N/A

**Metareview: Summary, Strengths And Weaknesses:**

This paper addressed the vision-language representation learning problem and proposed a distillation method to distill unimodal teacher models of vision and language onto vision-language (VL) student models for utilizing the benefit of unimodal pre-trained models. Reviewers have raised major weakness concerns on the incremental contributions, limited novelty, and flawed/weak experiments. Overall, the quality of this work is clearly below the acceptance bar and the authors did not provide any response to address the reviewers' concerns.